# Peroxiredoxins Play an Important Role in the Regulation of Immunity and Aging in *Drosophila*

**DOI:** 10.3390/antiox12081616

**Published:** 2023-08-15

**Authors:** Olena Odnokoz, Noah Earland, Marziyeh Badinloo, Vladimir I. Klichko, Judith Benes, William C. Orr, Svetlana N. Radyuk

**Affiliations:** Department of Biological Sciences, Southern Methodist University, Dallas, TX 75275, USA; olena.odnokoz@emory.edu (O.O.); earland.noah@wustl.edu (N.E.); m_badinloo@yahoo.com (M.B.); vklichko@smu.edu (V.I.K.); jbenes@smu.edu (J.B.); borr@smu.edu (W.C.O.)

**Keywords:** peroxiredoxin, reactive oxygen species, redox state, immunity, aging, mitochondria, endoplasmic reticulum, *Drosophila*

## Abstract

Aberrant immune responses and chronic inflammation can impose significant health risks and promote premature aging. Pro-inflammatory responses are largely mediated via reactive oxygen species (ROS) and reduction–oxidation reactions. A pivotal role in maintaining cellular redox homeostasis and the proper control of redox-sensitive signaling belongs to a family of antioxidant and redox-regulating thiol-related peroxidases designated as peroxiredoxins (Prx). Our recent studies in *Drosophila* have shown that Prxs play a critical role in aging and immunity. We identified two important ‘hubs’, the endoplasmic reticulum (ER) and mitochondria, where extracellular and intracellular stress signals are transformed into pro-inflammatory responses that are modulated by the activity of the Prxs residing in these cellular organelles. Here, we found that mitochondrial Prx activity in the intestinal epithelium is required to prevent the development of intestinal barrier dysfunction, which can drive systemic inflammation and premature aging. Using a redox-negative mutant, we demonstrated that Prx acts in a redox-dependent manner in regulating the age-related immune response. The hyperactive immune response observed in flies under-expressing mitochondrial Prxs is due to a response to abiotic signals but not to changes in the bacterial content. This hyperactive response, but not reduced lifespan phenotype, can be rescued by the ER-localized Prx.

## 1. Introduction

The immune system evolved to protect organisms from microbial invaders and other immune stresses. With age, immune responses tend to become aberrant, resulting in functional deficit or hyperfunction. Immune hyperfunction can lead to chronic inflammation, which is responsible for the development of various diseases and premature aging [1,2,3]. Although correlations between the overactivity of the immune system and physiological deficits are well established, the need for a mechanistic understanding of the causes that underlie the age-related dysregulation of immunity is a roadblock to efforts toward the development of interventions to promote a more vigorous healthspan.

Several lines of evidence underscore the importance of redox and redox-sensitive signaling in modulating various facets of the immune state. However, the mechanisms by which changes in the redox state are transmitted into cellular responses remain largely unknown.

Recently, there has been increasing interest in the hyperactive immune response in the absence of infection or so-called sterile inflammation [4], a phenomenon with a potential age-related component. There are different sources of pathogen-irrelevant inflammatory triggers known as damage-associated molecular patterns (DAMPs), which include reactive oxygen species (ROS), changes in redox, and redox-related damages, among others [5,6,7]. Proper ROS concentrations and redox balance are maintained by different antioxidant and redox-regulating factors. Among such factors are thiol-dependent peroxidases, also called peroxiredoxins (Prx), which play a pivotal role in regulating cellular redox homeostasis. Our studies in *Drosophila* revealed that the chronic overactivation of immunity and shifts to a pro-inflammatory state during aging appear to depend on the activity of some members of the peroxiredoxin gene family [8,9,10].

There are numerous studies indicating the role of Prxs in modulating immune-related signaling [8,9,10,11,12]. Prx1 and Prx2 can inhibit NF-κB activation [11]. *Drosophila* peroxiredoxin (dPrx) 5 negatively regulates immunity and has a beneficial effect on longevity [10]. dPrx4 can activate NF-κB and induce inflammation [9,12]. dPrx4 overexpression triggered an NF-κB-mediated pro-inflammatory response, and its overexpression, particularly in the fat body, had a marginal negative effect on life span (4–8% relative to control flies) [9].

Prxs function by sensing and scavenging ROS, such as hydrogen peroxide, and use these molecules to transmit redox-sensitive signals [13,14]. The major sources of cellular ROS are the endoplasmic reticulum (ER) and mitochondria, where reactive species, such as H_2_O_2_, are produced in the process of oxidative protein folding and during respiration [15,16].

In recent studies [8,9], we identified two important ‘hubs’, the ER and mitochondria, where extracellular and intracellular stress signals are transformed into cellular pro-inflammatory responses. We found that these responses are modulated by the activity of Prxs residing in these cellular organelles. We also found that the *Drosophila* NF-κB-like transcription factor Relish is a key regulator responsible for immune hyperactivity in old flies. These led us to suggest that the redox-sensitive changes in the ER and mitochondrial signaling pathways result in chronic inflammation and, thus, premature aging and that the Prxs residing in these subcellular compartments largely dictate proper signaling. Here, we extended these studies and investigated the relationship between redox, immunity, and aging using the *Drosophila* model, in which the components of the immune and redox systems are remarkably preserved, all having mammalian orthologs.

We investigated the effects of mitochondrial dPrxs in tissues where immune response genes are produced, in particular in the fat body and intestinal epithelium [17,18]. We also investigated the nature of the immune response triggers in dPrxs mutants and determined that dPrx acts in a redox-dependent manner in regulating the age-related immune response. Finally, we explored the cross-talk between the ER and mitochondria, the organelles where most ROS are generated and that are sites of the induction of immune effectors that occurs in the absence of pathogenic cues. We determined the role of dPrxs, residing in these organelles, in immune response and life span.

## 2. Materials and Methods

### 2.1. Fly Strains and Procedures

All mutant, transgenic, and enhancer fly lines were backcrossed into the *y w* reference strain background at least eight times. The *daughterless* (Da-GAL4, global expression) and S106-pSwitch-GAL4 (inducible fat-body-specific expression) driver lines were supplied by Dr. Blanka Rogina (University of Connecticut Health Science Center). The NP1-GAL4 (midgut-specific expression) driver [19] was a kind gift from Dr. Heinrich Jasper (Genentech). The characteristics and attributes of the drivers are outlined in FlyBase and have also been documented in publications [20,21]. The *dprx5* mutant allele is described in Michalak et al. [22]. Under-expression of dPrx3 and dPrx4 was achieved using UAS-RNAi-*dprx3* and UAS-RNAi-*dprx4* transgenic fly lines as described previously [12,23]. Under-expression of dPrx3 in the fat body, in the midgut, or globally was achieved by crossing the UAS-RNAi-*dprx3* transgene with the S106-pSwitch-GAL4, NP1-GAL4, and Da-GAL4 drivers, respectively. To induce the S106 pSwitch-GAL4 driver, flies were fed food containing drug RU486 (experimental) or drug diluent, ethanol (control), as described in [9]. Flies under-expressing both dPrx3 and dPrx5 were generated as described in [23]. Flies under-expressing dPrx3, dPrx4, and dPrx5 were generated by expressing the RNAi-*dprx3* and RNAi-*dprx4* transgenes, using the ubiquitous Da-GAL4 driver in the *dprx5*^−/−^ mutant background. Corresponding alleles are described in our previous publications [12,23]. The RNAi-*dprx4*; Da-GAL4, *dprx5*; and *dprx5*, RNAi-*dprx3* configurations were obtained by recombination. Table 1 presents the genotypes of flies and corresponding abbreviations. Survivorship studies were conducted as outlined in our previous publications [8,9,24,25]. Fly deaths were recorded approximately every 24 h.

### 2.2. Intestinal Barrier Dysfunction

Intestinal barrier dysfunction was tested using the smurf assay [26]. Briefly, flies were maintained under standard conditions until the day of the assay. On the assay day, flies were transferred to vials containing blue fly food medium. The blue medium was prepared by adding 2.5% blue dye no. 1 (Erioglaucine disodium salt, pure, ACROS Organics 3844-45-9, Geel, Belgium) to the standard medium (wt/vol). Flies were maintained on a blue medium until death. When blue coloration was observed outside the digestive tube, a fly was considered a smurf. The assay is based on the principle that the gut epithelia should absorb the blue food dye. In flies with normal intestinal function, the blue stain is limited to the intestinal lumen. However, if intestinal barrier dysfunction exists, the blue staining spreads throughout the body, resulting in the ‘smurf’ phenotype.

### 2.3. Bacterial Load Study

Three types of antibiotic food were used: tetracycline (final concentration 100 μg/mL), a combination of ampicillin and chloramphenicol (final concentration 50 μg/mL for each), and a combination of doxycycline and gentamycin (final concentration 50 μg/mL for each). All antibiotics were purchased from Sigma-Aldrich, USA. We selected this combination for its broad-spectrum antibacterial effect against both Gram-positive and Gram-negative bacteria. To prepare the antibiotic food, we melted standard fly food, cooled it to 50–60 °C, mixed it with the appropriate antibiotics, and poured the mixture into clean vials.

To assess the bacterial load, we extracted DNA from 12-day-old male flies using the Puregene Core Kit A (Qiagen, Germantown, MD, USA). Prior to DNA extraction, external bacteria were eliminated by washing the flies with 500 μL of 0.1% Tween20 (Sigma-Aldrich P7949, Saint Louis, MO, USA) and 500 μL of 70% ethanol. After removing the ethanol, the flies were homogenized for DNA extraction. The DNA concentration was measured and adjusted to 200 ng/μL using nuclease-free water. Subsequently, the 200 ng/μL DNA samples were diluted by adding 70 μL of nuclease-free water to 15 μL of the sample. The diluted samples were heated for 5 min at 95 °C. These DNA samples were used as templates for quantitative RT-PCR reactions using universal 16S rRNA primers: forward primer 27F 5′-agagtttgatcctggctcag-3′ and reverse primer 1492R 5′-acggctaccttgttacgactt-3′ (Thermo Fisher Scientific, Waltham, MA, USA). The quantitative PCR was performed on total genomic DNA to determine the ratio of bacterial DNA (16S rRNA) to fly DNA (ribosomal protein L32 (rp49)) in each sample.

### 2.4. DNA Preparation

We used a Genta Puregene Tissue Kit (Qiagen, Germantown, MD, USA) for DNA extraction and purification. Fifty adult flies were ground in 500 μL of cell lysis solution with a pestle. To digest proteins, 3 μL of proteinase K was added to the lysate, mixed by inverting 25 times, and incubated at 55 °C overnight. To degrade RNA, 4 μL of RNase A solution was added to the samples, mixed by inverting 25 times, and incubated at 37 °C for 1 h. After samples were cooled on ice for 3 min. Proteins were removed by adding 200 μL of protein precipitation solution, vortexing vigorously for 20 s, and incubating on ice for 2 min with subsequent centrifugation at 12,000× *g* for 5 min at 4 °C. The supernatant was mixed with phenol-chloroform (Sigma-Aldrich 77617, Saint Louis, MO, USA) in a 1:1 ratio and centrifuged at 12,000× *g* for 10 min at 4 °C. The upper phase was carefully poured into a microcentrifuge tube with 2-propanol (Fisher Scientific, Waltham, MA, USA) and mixed by inverting gently 50 times with subsequent centrifugation at 12,000× *g* for 10 min at 4 °C. The supernatant was carefully discarded, and the pellet was washed with 600 μL of 70% ethanol and centrifuged at 12,000× *g* for 1 min at 25 °C. This washing step was repeated with 100 μL of 70% ethanol. After removing the supernatant, tubes with pellets were kept open for 5 min. The DNA pellets were dissolved in 50 μL of DNA Hydration Solution overnight at 4 °C.

### 2.5. Quantitative RT-PCR and Immunoblot Analyses

Quantitative RT-PCR and Immunoblot analyses were performed as described [8]. Primer sequences were described in Odnokoz et al. [8]. Anti-dPrx3, dPrx5, and dPrx4 antibodies are described in our previous publications [12,23,27]. Anti-actin antibodies were purchased from MP Biomedicals, Solon, OH, USA.

Briefly, RNA was extracted from flies using the Trizol-chloroform method and adjusted to 100 ng/µL for reverse transcription reaction. The PCR cycling condition included an initial denaturation at 95 °C for 2 min followed by 45 cycles of 95 °C for 30 s, 55 °C for 30 s, and 72 °C for 30 s. Quantitative real-time PCR analysis was performed using Rotor-Gene™ RG-3000 (Corbett Research, Sydney, Australia) and software Q2.1.0.9. Signals obtained for a target gene were standardized against signals obtained for a house-keeping gene (rp49) in parallel sets of reactions. Expression of genes was calculated relative to the rp49 housekeeping gene using ΔΔCt method and represented as arbitrary mRNA units.

Anti-dPrx3, dPrx5, and dPrx4 antibodies were raised against recombinant dPrx proteins using services of the Proteintech Group, Inc. (Rosemont, IL, USA) and Covance Research Products (Denver, PA, USA) as described in our previous publications [12,23,27]. Antisera from the final bleeding were used at a 1:5000–1:10,000 dilution. Anti-actin antibodies were purchased from MP Biomedicals (MP Biomedicals, Solon, OH, USA) and used at 1:50,000 dilution.

### 2.6. Statistical Methods

Statistical analysis was performed using GraphPad Prism v10.0.2 and Excel Microsoft 365. Differences in protein, DNA, and mRNA levels were compared between groups by analysis of variance using Prism software v10.0.2 (GraphPad Software, Inc., La Jolla, CA, USA). Mean life spans and the statistical significance of survival curve comparisons were calculated using the log-rank test (Prism software v10.0.2). Differences were considered statistically significant at *p* < 0.05. Sample size and statistical methods are listed in figure legends.

## 3. Results

### 3.1. Life Span of Flies Under-Expressing Mitochondrial Peroxiredoxins in Intestinal Epithelium and Fat Bodies

We have previously found that targeting the under- or overexpression of dPrxs to different tissues can have differential effects on fly physiology [9,24]. We recently reported that the under-expression of mitochondrial dPrxs in motor neurons has a negative impact on life span comparable to global under-expression [24]. Here, we investigated the effects of mitochondrial dPrxs on tissues involved in *Drosophila* immunity, which largely relies on the function of fat bodies and intestinal epithelium or sites where a hallmark of *Drosophila* immunity, antimicrobial peptides (AMPs), are produced [20,28,29,30]. In the intestinal epithelium, AMPs are constitutively produced to maintain the homeostasis of commensal microbes. Fat bodies produce AMPs in an infection-inducible manner [31].

We investigated the effect of the depletion of mitochondrial dPrxs in the intestinal epithelium and fat bodies and how it contributes to the observed double mutant (DM) phenotype. To investigate the effect of the depletion of both mitochondrial dPrxs, we targeted the expression of the UAS-RNAi-*dprx3* transgene to the midgut and fat body using the NP1-GAL4 and S106-pSwitch-GAL4 drivers, respectively, in the *dprx5* mutant background. Despite the effective reduction in mitochondrial dPrx3 expression with the fat-body specific driver (Appendix A), there were no significant effects on longevity (Figure 1A). In contrast, the under-expression of dPrxs in the midgut (Appendix A) had a significant negative effect on life span, although this was not as pronounced as in the DM with the global under-expression of mitochondrial Prxs (Figure 1B).

### 3.2. Age-Dependent Changes in Intestinal Barrier Function in the Double and Single dPrx Mutants

Given the negative effect of the under-expression of mitochondrial dPrxs in the intestinal epithelia, we initiated efforts to identify specific factors involved in the life-span-shortening effect. We tested whether these effects could be due to intestinal barrier dysfunction.

The intestinal barrier serves as the first layer of defense to maintain organismal health, keeping the products of microbial metabolism away from the circulation, maintaining commensal bacteria interactions, and supporting healthy digestion [32,33,34]. Failure in the barrier might drive systemic inflammation and subsequent premature death. A decline in intestinal barrier function is a characteristic of aged flies and can be tested using the ‘smurf’ assay (Section 2, [26]), in which tissue integrity is measured by leakage of blue dye administered in the diet.

The ‘smurf’ phenotype (Appendix A) normally develops within approximately 24 h before death. Initially, the number of flies with the ‘smurf’ phenotype was counted in the double and single mutants under-expressing Prxs globally (see Table 1). One day before death, approximately 20% of the female DM flies and 15% of the male DM flies exhibited the ‘smurf’ phenotypes, which is 3–4 times less than the percentage of the ‘smurf’ flies in control and single mutants (Figure 2). While the appearance of the ‘smurf’ phenotype in the latter half of its life span seems to scale with that observed in control flies, the extent of its appearance is significantly lower and plateaus more rapidly. Thus, it would appear that, in the DM, rapid death tends to occur prior to the development of intestinal barrier dysfunction, and the intestinal barrier decline is not the only factor contributing to the rapid death of the DM.

Since only a small percentage of the short-lived DM flies under-expressing mitochondrial dPrxs globally developed intestinal barrier dysfunction before death (Figure 2), we decided to conduct the study in the intestinal epithelial-specific DM flies (NP1 DM). Almost all NP1 DM and NP1 control flies exhibited intestinal barrier dysfunction a few days before death because the overall mortality and ‘smurf’ curves matched (Figure 3). These results suggest that, while intestinal barrier dysfunction is not contributing to the rapid death in the DM flies under-expressing Prxs globally, mitochondrial dPrxs in the intestinal epithelium are essential for maintaining normal intestinal barrier functioning.

### 3.3. The Overactivation of AMPs in the DM Was Not Due to Changes in Microbial Load

Second, we wanted to explore more the nature of factors that drive the activation of the immune system during aging. Since we found that under-expression of mitochondrial dPrxs is sufficient to cause the up-regulation of the AMPs [8], we wanted to determine to what extent these factors are of microbial origin or to what extent their presence is microbe-irrelevant.

Normal fly aging is characterized by hyperactivation of immunity and increased bacterial load [35]. The mitochondrial DM flies under-expressing dPrxs globally exhibited a chronically hyperactive immune response [23]. To determine whether this was due to a response to changes in microbial content or abiotic factors, we examined total bacterial load in the DM and control flies and in the DM flies fed antibiotics. We used different classes of broad-spectrum antibiotics and their combinations as they are effective towards different microbial species.

Bacterial loads were examined in control and DM flies of the same chronological age (~12 days) or at the time point when DM flies showed 10% mortality in their life trajectory. The study showed that the DM flies had a higher total bacterial load compared to control flies, which was significantly reduced by antibiotic activity (Figure 4A).

However, this reduction in microbial load had little or no effect on the levels of AMPs (Figure 4B) and the DM flies’ life span (Figure 4C). Altogether, the data suggest that the induction of AMPs in the DM is a response to abiotic signals rather than a response to changes in the microbiota. These signals have a mitochondrial origin and correlate with the activity of mitochondria-localized dPrxs.

### 3.4. dPrx5 Suppresses Hyperactive Immunity in Old Flies Due to Its Peroxidase Activitye

Previously, we showed that the under-expression of mitochondrial dPrx3 and dPrx5 individually or together (DM) cause the age-dependent changes in AMP levels similar to those observed in controls when scaled to the percentage of life span [8]. We also found that flies lacking dPrx5 were more resistant to bacteria, eliciting a more robust up-regulation of the immunity genes. In contrast, higher levels of dPrx5 conferred greater susceptibility to infection and a dampened activation of humoral immunity [10]. Here, we investigated whether dPrx5 might have similar effects on the hyperimmune response associated with aging.

Consistently with previous reports, the AMP levels in the *dprx5* mutant were only moderately higher in old flies compared to the controls (Figure 5). In contrast, the high-level global overexpression of dPrx5 significantly suppressed the age-related hyperactivity of AMPs. The data suggest that pathways that regulate the activation of AMPs in response to infection may be partially shared with those that regulate the AMPs in response to unknown age-related signals. At the same time, when the redox-negative (RN) form of dPrx5 was expressed [10], no AMP suppression was detected (Figure 5), which indicates that, as in response to infection, dPrx5 acts in a redox-dependent manner in regulating the age-related immune response.

### 3.5. Interactions between the ER and Mitochondrion-Mediated Signaling in Eliciting an Age-Related Chronic Inflammatory Response

The data obtained in mammals point to a close cooperation between the ER- and mitochondria-originated signaling pathways in regulating pro-inflammatory response [36,37]. Both organelles are major sources of cellular ROS. ROS affect the signaling processes in these two organelles, and they are also involved in mediating intercommunication. ROS levels and ROS-mediated cellular processes should be under tight control to ensure proper cellular functioning, and the necessity for such control underscores the importance of Prxs.

Previously, we found that the depletion of ER- and mitochondria-localized dPrxs impact the state of humoral immunity in flies and share common signatures of the inflammatory response, particularly the comparable activation of Relish-dependent AMPs [9,10,23]. Furthermore, a similar activation is observed in flies undergoing normal aging (Figure 5). We also found that the activity of the ER-specific dPrx4 is required for inflammatory/immune responses to oxidants, such as paraquat [9]. Based on the assumption that signals due to paraquat exposure could mimic signals due to changes in mitochondrial redox, we investigated whether the effects of mitochondrial dPrxs are mediated via the ER-mitochondrial cross-talk. For that purpose, we used a triple mutant under-expressing dPrx3 and dPrx5 (mitochondrial) and dPrx4 (ER-localized) (see Section 2) where under-expression of these dPrxs was confirmed by immunoblot analysis (Appendix A).

The results showed that the activation of the immune response, observed in the mitochondrial DM, was disrupted by the simultaneous under-expression of the ER-localized dPrx4 (Figure 6B,C), suggesting that the mitochondria-originated non-pathogen activation of immunity proceeds via the ER pathway and is mediated via its resident dPrx4. However, the dramatic reduction in the life span of the DM was not rescued when dPrx4 was simultaneously under-expressed by RNAi (Figure 6A and Appendix A). The global under-expression of dPrx4 by RNAi alone had little or no effect on life span, with mean life span being similar to control, as shown previously [9,12]. Although there were significant differences between DM and TM flies in some experiments, they displayed sexual disparities (Appendix A). Additionally, these differences were marginal compared to differences between the life spans of flies under-expressing dPrx4 alone or control and triple mutant flies. Furthermore, these disparities were not replicated in other experiments (Figure 6A), and the differences in life spans between DM and TM were not significant. Therefore, we came to the conclusion that the longevity phenotype of the triple mutant was mainly determined by the reduced activity of mitochondrial dPrxs. Thus, the obtained data suggest that the induction of AMPs in response to abiotic signals is not a critical determining factor of aging and that the observed shortening of life span in the mitochondrial DM is due to other factors but not overactive immunity.

## 4. Discussion

In previous studies [8,23,24], we found that flies under-expressing mitochondrial Prxs globally or in neuronal tissues developed characteristics of rapid physiological aging. Here, we extended the study to pinpoint other tissues that might affect aging in response to the depletion of mitochondrial Prx activity and found a significant effect on the survivorship of flies under-expressing mitochondrial dPrxs in the intestinal epithelia but not fat bodies (Figure 1).

The fat body is a tissue that mediates the immune response in *Drosophila* and where immunity-related genes (IRG) are produced in response to systemic infections, as well as during aging due to unknown stimuli [18,38,39,40,41,42,43,44,45]. The *Drosophila* fat body, the fly equivalent of adipose tissue and liver in mammals, is one of the major sites of inducible AMP production and has been shown to play an important role in modulating longevity [29,46]. Surprisingly, the fat-body-specific under-expression of mitochondrial dPrxs had little to no effect on the life span (Figure 1A). One plausible explanation for this finding is that the fat body tissue is more resistant to oxidative insult caused by mitochondrial dPrxs under-expression. Indeed, in our previous study, we found increased apoptosis in the cardia, muscles, and gut epithelia but not fat bodies [23].

In contrast to the fat body, the effects of the under-expression of mitochondrial dPrxs had a significant impact on the functioning of the fly mucosal immune organ, more specifically, the intestinal epithelium of the midgut (Figure 1B). In addition to serving as a physical barrier, intestinal epithelial cells are an important facet of immunity as part of the mucosal immune system and one of the major sites of production of immune-related genes (IRG) [18,33,43]. Like many other organs, the gastrointestinal tract is also a site of heightened oxidative stress during aging. Thus, the targeted under-expression of mitochondrial dPrxs in the gut seemed to have the accelerated onset of intestinal barrier dysfunction. Presumably, mitochondrial dPrxs are essential for ‘proper’ ROS control or may target yet-to-be-determined redox-sensitive pathways.

It is well known that the aging process is associated with the deterioration of the gastrointestinal tract. Zheng et al. showed that old organisms have increased levels of apoptotic cells in intestinal epithelium [47]. Our studies revealed similar changes in the intestinal epithelium in short-lived flies under-expressing mitochondrial dPrxs, suggesting a functional decline in this tissue due to a reduction in mitochondrial Prx activity [23]. This was in agreement with the data obtained by Biteau et al., where flies with accelerated intestinal dysplasia were shown to have a short life span [48].

In addition to the increase in the incidence of apoptosis observed in the previous study [23], here, we found other indicators of the breached integrity of the intestinal epithelium. Thus, the depletion of mitochondrial dPrxs using the midgut-specific driver NP1 led to signs of intestinal barrier dysfunction, as evidenced by the appearance of the ‘smurf’ phenotype in most dead flies (Figure 3). The ‘smurf’ phenotype indicates intestinal dysfunction and has been used as a signature of impending mortality in various fly populations [26].

As an indicator of the loss of intestinal integrity, the ‘smurf’ phenotype has been reported to develop in flies undergoing aging. It is thought to reflect the penetration of microbes and toxins as the barrier deteriorates [26]. It has been shown that the development of this phenotype depends on ROS and the redox state of the cells of intestinal tissue, where higher ROS may lead to inflammation and cell death [32,34,49,50,51]. Thus, specific targeting to intestinal tissues had a marked effect on the development of this phenotype, and mortality was due solely to the impact on the intestinal epithelium (Figure 3). It was not the case when mitochondrial dPrxs were under-expressed globally (with the Da driver, DM) as the frequency of ‘smurf’ was markedly different from the mortality rate (Figure 2). This suggests that other organs critical for aging were affected and that these mutants die before developing gut barrier dysfunction due to other factors contributing to mortality.

Another sign of loss of gut integrity was a significant increase in bacterial load in DM flies compared to control flies of the same chronological age (Figure 4A), which we examined as age-related gut barrier dysfunction ascribed to increased bacterial load and AMP induction [26]. The microbial load in the intestine is mainly controlled by two factors, ROS and AMPs, which act in parallel but are mutually dependent (reviewed in Kim [7]).

Conversely, since an increase in the bacterial load and hyperactivation of immunity in *Drosophila* are characteristics of ‘normal’ aging [35,52,53], it has been suggested that bacteria can enhance the age-related induction of immunity-related genes. However, studies that showed that IRG genes were still significantly induced in old flies maintained in a sterile environment or with antibiotics indicate that IRGs respond to signals not directly associated with microbes [38,40]. We also observed a significant induction of AMPs in physiologically old DM flies regardless of the presence of antibiotics (Figure 4B), suggesting a response to abiotic stimuli rather than to intestinal microbial content. Thus, another important finding of our study is that the development of hyperactive humoral immunity in the DM is a response to abiotic signals that likely originate in mitochondria rather than increase in the microbial content. Our previous finding that the age-related increase in AMP levels is not triggered via canonical immune pathways but responds to age-related endogenous changes, such as increased ROS production by mitochondria, supports this view [54]. However, these changes in microbial load did not cause any effects on life span (Figure 4C). We have yet to explore the potential effects on the commensal microbiota and changes in the gut microbiome that are implicated in healthy intestinal functioning and reduced pro-inflammatory signaling [55].

We also determined that the ability of dPrx5 to suppress the age-related overproduction of AMPs is entirely dependent on its peroxidase activity (Figure 5), supporting the notion that age-related AMP activation depends on redox signals. This can be explained by a dual role of ROS. In addition to directly killing microbes, they regulate AMP production systemically and might be responsible for AMP up-regulation observed in the DM and old controls [56]. Considering our previous findings that the global overexpression of dPrx5 promotes longevity by its antioxidant and anti-apoptotic activities [27], the data from this study suggest that dPrx5 may also have effects on aging by suppressing age-related AMP activation. However, the causal relationship is still to be established.

Inflammation in *Drosophila* typically results in the production of AMPs [57]. ROS and changes in redox are among the triggers of sterile inflammation [7,31,56]. The major sources of these triggers are the ER and mitochondria [15,16], both of which are regarded as important factors in the developing non-pathogen-related inflammation [9,15,36,58]. The concentration of these species is maintained by Prxs, residents of these organelles, the activity of which is responsible for modulating the immune response, as shown in our previous studies [8,12,23]. Our data obtained with a triple mutant lacking the activity of dPrx localized in the ER and mitochondria indicate that redox-dependent signaling pathways originating in these organelles interact in controlling the state of immunity and inflammation. The data obtained in mammals also point to a close cooperation between the ER- and mitochondrial signaling pathways in the regulation of the pro-inflammatory response [36,37].

To conclude, our data demonstrated that the abiotic signal that induces an immune response arises in mitochondria with impaired redox due to the reduced activity of resident dPrxs. This signal is transmitted through the ER and requires the activity of the ER-localized dPrx4, which culminates in AMP induction (Figure 6). However, the decrease in AMP levels in the triple mutant was not associated with life span, which remained shortened, indicating that other factors play a causal role in the shortened longevity of mutants with reduced mitochondrial Prx activity.

## Figures and Tables

**Figure 1 antioxidants-12-01616-f001:**
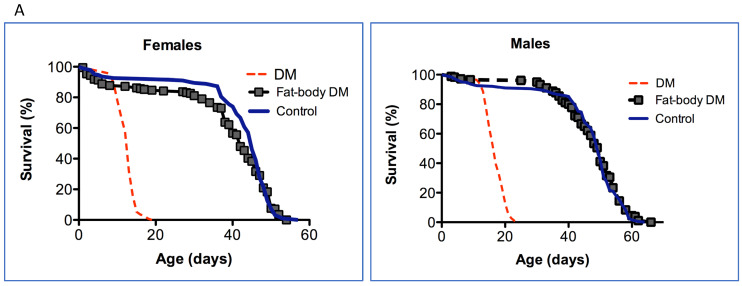
The effect of under-expression of mitochondrial dPrxs in the fat body (**A**) and in the midgut (**B**) on the life span of flies. dPrx3 was under-expressed in the *dprx5* null background using the inducible S106-pSwitch-GAL4 (fat body) and NP1-GAL4 (midgut) drivers. For comparison, dPrx3 was under-expressed globally with the Da-Gal4 driver (DM, red dotted lines). (**A**) To activate the gene switch-inducible driver, S106 DM flies were fed food containing RU486 (experimental, fat-body DM), while control flies were fed food with ethanol (control), as described in Section 2. (**B**) DM—Double mutant under-expressing dPrx3 globally with Da-GAL4 driver; NP1 DM—double mutant under-expressing dPrx3 in the midgut; Control—NP1-GAL4 driver. Approximately 100–125 flies were used for each fly line in the experiment. Shown are representative data of two independent biological experiments, and similar results were obtained in a replicate experiment. A summary of the data is presented in Table 2. The fly lines and genotypes of the flies are described in detail in Table 1.

**Figure 2 antioxidants-12-01616-f002:**
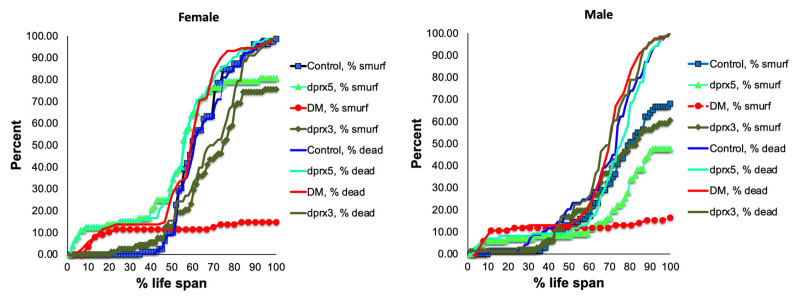
Effects of global under-expression of mitochondrial dPrxs on survivorship and development of the ‘smurf’ phenotype. The measurements were performed in the control, DM, and single (*dprx3* and *dprx5*) mutant female (**left**) and male (**right**) flies (n = 125 for each group of flies). A number of flies with the ‘smurf’ phenotype were counted after feeding the flies with food containing the blue dye added to food. The percentage of dead and ‘smurf’ flies were normalized to the percent of life span for comparison between mutant and control flies. Statistically significant difference (*p* < 0.05) was observed between percent of DM total dead flies (DM, % dead) and DM dead flies that developed the ‘smurf’ phenotype (DM, % smurf). Differences between dead and ‘smurf’ flies in single dPrx mutants and control were not significant. Shown are representative data of two independent biological replicates. The results of biological replicate experiment are shown in Appendix A. The names of fly lines and genotypes of flies are described in Table 1.

**Figure 3 antioxidants-12-01616-f003:**
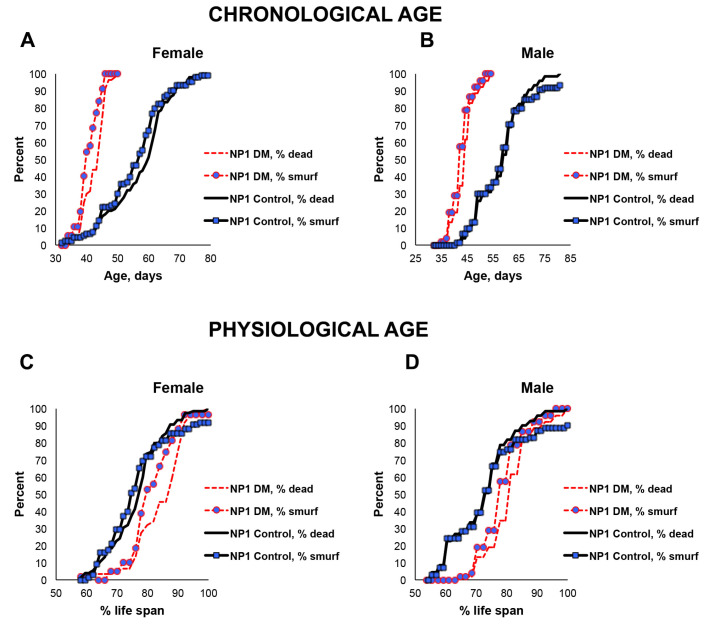
Effects of under-expression of mitochondrial dPrxs in the midgut on survivorship and development of the ‘smurf’ phenotype. The measurements were performed in the NP1 DM and NP1 control female (**A**,**C**) and male (**B**,**D**) flies (n = 125). A number of flies with the ‘smurf’ phenotype were counted after feeding the flies with food containing the blue dye added to food. Percentage of the dead and ‘smurf’ flies is shown on the *y* axis as a function of physiological aging (% of life span, *x* axis). Shown are representative data of two independent biological replicates. Approximately 100–125 flies were used for each fly line. The results of the biological replicate experiment are shown in Appendix A. Percentage of dead and ‘smurf’ flies were scaled to chronological age (**A**,**B**) and normalized to percent of life span or physiological age (**C**,**D**). There was no statistically significant difference between dead and ‘smurf’ flies (*p* > 0.05). The names of fly lines and genotypes of flies are described in Table 1.

**Figure 4 antioxidants-12-01616-f004:**
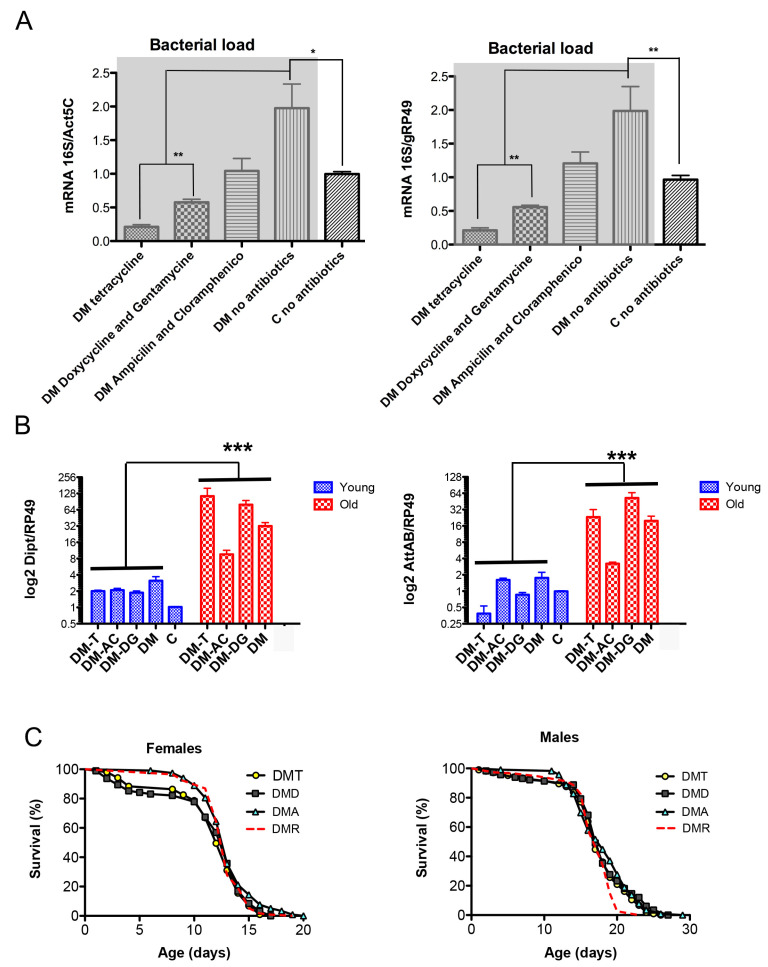
Effects of antibiotics on bacterial load (**A**), expression of AMPs (**B**), and life span (**C**) of the DM flies. Flies were maintained on regular food or food supplemented with tetracycline (T), a combination of ampicillin and chloramphenicol (AC), or a combination of doxycycline and gentamicin (DG). (**A**) Genomic DNA was isolated from 12-day-old DM and control (C) flies, and total bacterial load was determined using universal 16S rRNA primers. Results are means ± SEM of three replicates performed with two independent cohorts of flies (total n = 6). Results for DM flies with and without antibiotics are highlighted in gray. Asterisks denote statistically significant differences obtained in qPCR analysis (* *p* < 0.05; ** *p* < 0.005). (**B**) RNA was isolated from DM flies of two ages, 5–6 days (physiologically young) and 14–15 days (physiologically old, about 10% of fly death), and from 14-day old control flies (+/Da, *dprx5*, C), considered physiologically young. Results are means ± SEM of three replicates performed with two independent cohorts of flies (total n = 6). The two-way ANOVA test showed significant differences between physiologically young and old age (*** *p* < 0.0001). (**C**) Shown are data from one experiment with approximately 100–125 flies for each line. The results of the biological replicate from an independent cohort are shown in Appendix A. Statistical data are shown in Appendix A. The DMs were kept on the standard food (DMR) and food supplemented with tetracycline (DMT), a combination of ampicillin and chloramphenicol (DMA), or a combination of doxycycline and gentamicin (DMD). The log-rank test did not show significant differences between fly groups.

**Figure 5 antioxidants-12-01616-f005:**
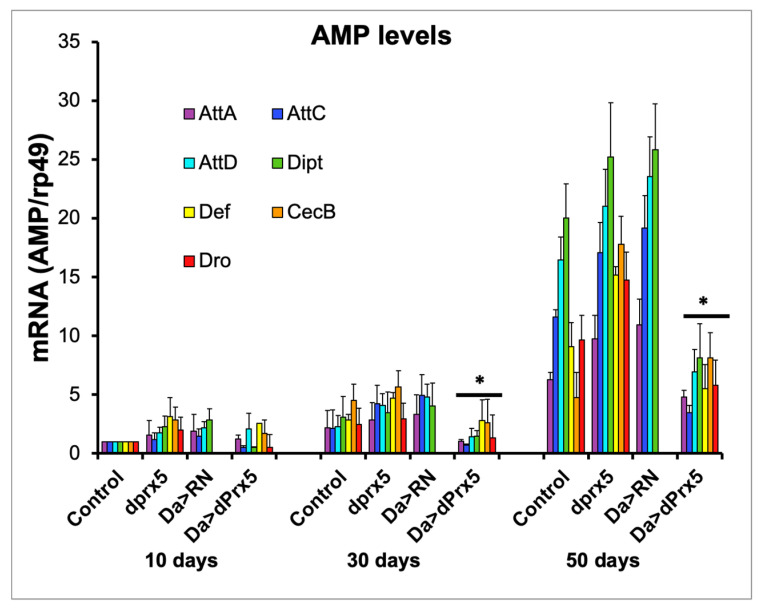
AMP expression in young (10 days), middle-aged (30 days), and old (50 days) flies. AMPs tested were attacins (*AttA*, *AttC*, and *AttD*); defensin (*Def*); Diptericin (*Dipt*); drosocin (*Dro*); and cecropin C (*CecC*). Level of AMPs in driver control flies, *dprx5* mutants (dprx5); and in flies expressing the dPrx5 transgene (Da > dPrx5) or its RN form (Da > RN) in the endogenous gene null background. Signals for each AMP were standardized against signals obtained for rp49 housekeeping gene and plotted on *y* axis. Results are means ± SEM of three replicates performed with two independent cohorts of flies (total n = 6). Asterisks denote statistically significant differences obtained in RT-PCR analysis between AMP levels in flies overexpressing wild type dPrx5 transgene and flies overexpressing the redox-negative form of dPrx5, *dprx5* mutants, and control (* *p* < 0.05).

**Figure 6 antioxidants-12-01616-f006:**
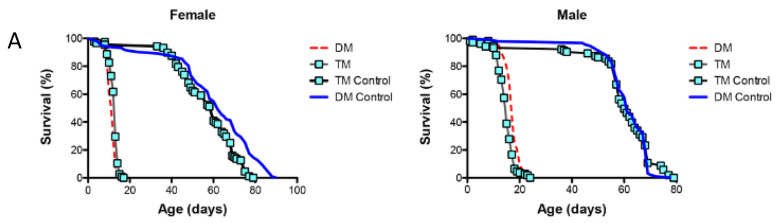
The role of ER and mitochondrial components in age-dependent induction of the immune-related genes. (**A**) Survivorship curves for females (**left**) and males (**right**). Shown are data from one experiment with approximately 100–125 flies for each line. The results of a biological replicate from an independent cohort are shown in Appendix A. Statistically significant differences (*p* < 0.05) between survivorship curves were determined by the log-rank test (Table 3). (**B**) RT-PCR analysis of AMP expression in dPrx double and triple mutants at different ages. The age-dependent changes in *Diptericin* (**left**) and *AttAB* (**right**) expression levels. The genotypes of flies are described in Table 1. All groups of flies were collected at different ages, as indicated in Appendix A. Results are means ± SEM of two replicates performed with three independent cohorts of female flies (total n = 6). The statistically significant differences in age-specific changes in the levels of *Dipt* and *AttAB* between the DM, TM, and corresponding controls were determined by analysis of the slopes of corresponding regression lines (Table 4). (**C**) RT-PCR analysis of *Diptericin* expression in dPrx male mutants at different ages. **Left**: changes in mRNA levels in the double *dprx3*,*dprx5* mutant (DM), *dprx4*,*dprx3*,*dprx5* triple mutant (TM), and Da driver control; **right**: changes are scaled to physiological aging displayed as % of life span. Shown are representative data from one experiment with male flies (n = 3).

**Table 1 antioxidants-12-01616-t001:** Genotypes of control and experimental flies and fly line names.

**Genotypes and Abbreviations of Transgenic Lines**		
**Line Name**	**Genotype**	**Abbreviation**
*dprx5* null mutant	*dprx5*/Da-GAL4, *dprx5*	*dprx5*
RNAi-*dprx3* transgenic flies	RNAi-*dprx3*/Da-GAL4, *dprx5*	*dprx3*
Double mutant	RNAi-*dprx3*, *dprx5*/Da-GAL4, *dprx5*	DM
Control or double-mutant control	*+*/Da-GAL4, *dprx5*	Control or DM control
Triple mutant	RNAi-*dprx4*/+; RNAi-*dprx3*, *dprx5*/Da-GAL4, *dprx5*	TM
Triple-mutant control	RNAi-*dprx4*/+; RNAi-*dprx3*, *dprx5*/+	TM control
**Genotypes and Abbreviations of Flies with Tissue-Specific under Expression of** **dPrx3 in *dprx5* Null Mutant Background**		
**Line Name**	**Genotype**	**Abbreviation**
Intestinal DM	*+/+;* NP1-GAL4/+; RNAi-*dprx3*, *dprx5*/*dprx5*	NP1 DM
Intestinal control	+/+; NP1-GAL4/+; *dprx5*/*+*	NP1 control
Fat-body DM (with mifepristone 100 μg/mL)	+/+; S106-pSwitch-GAL4/+; RNAi-*dprx3*, *dprx5*/*dprx5*	S106 DM
Fat-body control (with ethanol, diluent of mifepristone)	+/+; S106-pSwitch-GAL4/+; RNAi*-dprx3, dprx5/dprx5*	S106 control

**Table 2 antioxidants-12-01616-t002:** The mean life span of the DM flies under-expressing dPrx3 in the fat body and midgut in the *dprx5* null mutant background shown in Figure 1. Columns 1 and 4 indicate the mean life spans observed in two independent biological experiments. Columns 2 and 5 display the percentage changes in the experimental groups compared to their corresponding controls. Columns 3 and 6 indicate the significance probabilities obtained from the log-rank tests, which were used to compare the survival curve. Statistically significant differences are indicated in bold.

Females				Males			
	1	2	3		4	5	6
Line	Mean, Days	% vs. Control	*p*-Value	Line	Mean, Days	% vs. Control	*p*-Value
S106 Control	45			S106 Control	49		
	44				40		
S106 DM	42	−6.67	>0.05	S106 DM	50	+2.00	>0.05
	37	−15.91	>0.05		40	0.00	>0.05
NP1 Control	58			NP1 Control	62		
	63				67		
NP1 DM	47	**−18.97**	**<0.0001**	NP1 DM	40	**−35.48**	**<0.0001**
	41	**−34.92**	**<0.0001**		51	**−23.88**	**<0.0001**
DM	15	**−74.14**	**<0.0001**	DM	19	**−69.35**	**<0.0001**
	13	**−79.37**	**<0.0001**		14	**−79.37**	**<0.0001**

**Table 3 antioxidants-12-01616-t003:** The mean life span of the triple- and double-mutant flies shown in Figure 6A. Column 1 indicates the mean life spans observed in two independent biological experiments. Columns 2 and 4 display the percentage changes in the experimental groups compared to the TM control and the DM, respectively. Columns 3 and 5 indicate the significance probabilities obtained from the log-rank tests, which were used to compare the survival curve. Statistically significant differences are indicated in bold.

	1	2	3	4	5
Line	Mean, Days	% vs. TM Control	*p*-Value	% vs. DM	*p*-Value
**Females**					
DM	12				
	14				
TM	13	**−78.33**	**<0.0001**	**+8.33**	**<0.0001**
	14			0.00	*p* > 0.05
DM Control	62	+3.33	*p* > 0.05	**+416.67**	**<0.0001**
TM Control	60				
**Males**					
DM	17				
	13.5				
TM	15	**−75.00**	**<0.0001**	**−11.76**	**<0.0001**
	13			**−3.70**	**<0.0001**
DM Control	61	+1.67	*p* > 0.05	**+258.82**	**<0.0001**
TM Control	60				

**Table 4 antioxidants-12-01616-t004:** Statistical analysis of age-dependent changes in AMP expression between TM, DM, and their corresponding controls depicted in Figure 6C. To compare the difference in trajectories of these changes during physiological aging, we performed comparison of regression curve slopes and intercepts using Prizm GraphPad Software v10.0.2.

Line	Slope, *p*-Value	Significance	Intercepts, *p*-Value	Significance
**Dipt**				
DM vs. TM	0.0038	Very significant		
TM vs. TM Control	<0.0001	Extremely significant		
DM vs. DM Control	0.595	Not significant	0.8306	Not significant
TM Control vs. DM Control	0.3628	Not significant	0.4407	Not significant
**AttAB**				
DM vs. TM	0.024	Significant		
TM vs. TM Control	0.0098	Very significant		
DM vs. DM Control	0.0030	Very significant		
TM Control vs. DM Control	0.0076	Very significant		

## Data Availability

The raw data supporting the conclusions of this article will be made available by the authors, without undue reservation.

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
