# Peer review of "Peroxiredoxins Play an Important Role in the Regulation of Immunity and Aging in Drosophila"

_antioxidants, 2023, doi:10.3390/antiox12081616_

Round 1
Reviewer 1 Report
Olena Odnokoz et al. reported data on the consequence of mutation of the mitochondrial peroxiredoxins on age-related immune responses in the drosophila fat body and intestinal epithelium.
The subject is of great interest and drosophila model is highly appropriate to decipher the role of peroxiredoxins on immune system and aging. The methodology is appropriate, and the data well described and support the conclusions.
Minor points:
Methods:
- For immunoblotting, the antibodies suppliers and used dilutions should be described.
- For RT-qPCR, quantity of RNA used for RT, the cyclin for qPCR and the calculation method for fold induction.
Results:
- Figures 5 and 6: the quality and the size of these figures are not appropriate and should be upgraded.
Author Response
We are very grateful to the reviewers for their constructive comments and suggestions. We have revised the manuscript based on the comments of reviewers and provided more clarifications, as requested. Our responses to the specific comments of the reviewers are given below. For convenience, all added text and changes are highlighted in yellow.
Reviewer 1:
Question:
Methods:
- For immunoblotting, the antibodies suppliers and used dilutions should be described.
- For RT-qPCR, quantity of RNA used for RT, the cyclin for qPCR and the calculation method for fold induction.
Answer: We have added a more detailed description of these methods in Materials and Methods.
Results:
- Figures 5 and 6: the quality and the size of these figures are not appropriate and should be upgraded.
Answer: We apologies for the inappropriate formatting. Now the figures are improved.
Reviewer 2 Report
This is a very interesting study that provides some support for the role played by peroxiredoxins in the regulation of the immune response in Drosophila.
I only have a few comments:
1. I think more evidence/data is required before the role of peroxirodoxins in the aging of Drosophila can be confirmed in view of the limited effect of AMP down regulation.
2. The final paragraph of the conclusion should be changed as the data only indicates that ` the abiotic signal that induces an immune response arises in mitochondrion with impaired redox due to reduced activity of resident dPrxs and that the signal is transmitted through the ER.
Author Response
We are very grateful to the reviewers for their constructive comments and suggestions. We have revised the manuscript based on the comments of reviewers and provided more clarifications, as requested. Our responses to the specific comments of the reviewers are given below. For convenience, all added text and changes are highlighted in yellow.
Reviewer 2:
Q1. I think more evidence/data is required before the role of peroxirodoxins in the aging of Drosophila can be confirmed in view of the limited effect of AMP down regulation.
Answer: We discussed the role of peroxiredoxins and AMPs in aging in more detail and included this in the Discussion section.
- The final paragraph of the conclusion should be changed as the data only indicates that ` the abiotic signal that induces an immune response arises in mitochondrion with impaired redox due to reduced activity of resident dPrxs and that the signal is transmitted through the ER.
Answer: We agree with the reviewer that the conclusion was exaggerated. We have changed the last paragraph in the Discussion accordingly.